# The role of fragrance and self-esteem in perception of body odors and impressions of others

Ilja Croijmans[1]*, Daniel Beetsma[1], Henk Aarts[1], Ilse Gortemaker[2,3], Monique Smeets[1,2]

**1** Department of Psychology, Utrecht University, Utrecht, The Netherlands, **2** UNILEVER R&D Beauty & Personal Care Science & Technology, Consumer Science, Rotterdam, The Netherlands, **3** ABN AMRO Bank N.V., Amsterdam, The Netherlands

* i.m.croijmans@uu.nl

## Abstract

Human sweat odor serves as social communication signal for a person's traits and emotional states. This study explored whether body odors can also communicate information about one's self-esteem, and the role of applied fragrance in this relationship. Female participants were asked to rate self-esteem and attractiveness of different male contestants of a dating show, while being exposed to male participant's body odors differing in self-esteem. High self-esteem sweat was rated more pleasant and less intense than low self-esteem sweat. However, there was no difference in perceived self-esteem and attractiveness of male contestants in videos, hence explicit differences in body odor did not transfer to judgments of related person characteristics. When the body odor was fragranced using a fragranced body spray, male contestants were rated as having higher self-esteem and being more attractive. The finding that body odors from male participants differing in self-esteem are rated differently and can be discriminated suggests self-esteem has distinct perceivable olfactory features, but the remaining findings imply that only fragrance affect the psychological impression someone makes. These findings are discussed in the context of the role of body odor and fragrance in human perception and social communication.

## Introduction

The ability to signal another person's non-verbal cues is considered functional social behavior. For instance, an individual's emotional expressivity may signal cooperative behavior and trustworthiness [1], and therefore receives more valuable social investments which in turn can benefit the social group at large on the longer term [2]. Information acquired via non-verbal communication promotes a more effective decision-making process by including whether these expensive individual investments in others are warranted [3]. This phenomenon is illustrated by the observation that people can judge personality traits predictive of future behavior from a face on first sight [e.g., 4–6], the tone of their voice [e.g., 7], or by the feel of a handshake [e.g., 8].

**Data Availability Statement:** https://osf.io/xwtnq/.

**Funding:** Unilever provided financial support for supplies and participant remuneration and material support by supplying fragranced (test product)

body sprays in cans free of labels. Unilever also provided support in the form of salaries for IG and MS. The specific roles of these authors are articulated in the 'author contributions' section. IG served as operations manager overseeing the project from Unilever. She provided feedback on study design and a manuscript draft. However, Unilever did not have a role in the data collection and analysis or decision to publish.

**Competing interests:** The authors have read the journal's policy and have the following competing interests: Unilever provided financial support for supplies and participant remuneration and material support by supplying fragranced (test product) body sprays in cans free of labels. IG and MS are employees of Unilever. This does not alter the adherence to PLOS ONE policies on sharing data and materials. A Unilever marketed product was used as body spray in the fragranced condition, but without a label or further identification. There are no patents, products in development, or other marketed products associated with this research to declare.

One important desirable social trait that fosters human interaction and cooperation pertains to people's self-esteem [9,10]. Self-esteem is defined as a person's appraisal of their value [10], and is often understood as *sociometer*: someone's self-esteem is dependent on how they think other members of a group they belong to, value them, and thus critically depend on their relationships [11,12]. It is regarded as a basic social-emotional trait, having a more-or-less stable trajectory through life, with self-esteem increasing somewhat with age [13]. People with high (vs. low) self-esteem have been shown to be happier, feel more confident, and are more liked and successful in social life [13], and low self-esteem is linked to feelings of loneliness and depression [14,15]. In addition, perceptions of self-esteem and associated psychological features can be inferred from others' appearance, facial expression or tone of voice [e.g., 6,7].

Recent research on the sense of smell shows that humans are sensitive to, and rely on, body odors for social communication. This so-called *social chemosignaling* [cf. 16], allows people, for example, not only to spot the tell-tale signs of inflammation from looking at a picture of an individual, but also to detect illness from the smell that person secretes [17]. In a similar vein, people can infer stable trait characteristics such as gender and relational status [e.g., 18,19], and personality traits such as dominance and extraversion from body odors [20]. In addition, body odors enable people to gauge more temporary emotional states, such as fear [e.g., 21,22], disgust [21], and anxiety [23]. Body odors may act as a signal, sent as a means for communication on purpose (e.g., when applying a perfume to make a particular impression when on a date), or as a cue that is understood in a particular way up by the perceiver, but not purposefully sent, (i.e., when someone gets a certain impression from someone else, by their particular body odor that is not directly manipulated [cf. 24]). Here, we investigate whether differences in self-esteem may lead to specific cues in body odors.

Differences in self-esteem could affect body odor in various ways. The trait might influence body odor in that high or low levels of self-esteem are associated with metabolic changes, which in turn are secreted onto the skin and result in different odors emanating from the body. Alternatively, people with differing levels of self-esteem might demonstrate typical behavior, e.g., they might interact with people more or less frequently, perform other types of sport, or have different hygiene routines, which might affect the body microbiome and in turn affect ones' body odor [25,26]. Another alternative route in which self-esteem *coincides* with a different body odor is that a different response of the HPA axis in potentially stressful situations affects the composition of sweat [e.g., 27,28], and thus, ones' body odor. In this latter mechanism, more general neurobiological mechanisms related to stress may affect self-esteem as well as body odor, such that body odor changes may coincide with both stress and self-esteem changes. This suggests that various mechanisms by which self-esteem might affect body odor of a sender are conceivable, and that these body odors may be picked up by perceivers through learning mechanisms, as having a communicative value about ones' personality.

Here we hypothesize that body odor can carry information about a person's self-esteem that can affect explicit social impressions, such as perceived personal attractiveness, acting as a *cue*. An important contribution of the present study therefore lies in exploring whether individuals with a low vs. high self-esteem secrete body odors that are distinctive and can influence subsequent impression formation. If such an effect would emerge then this provides crucial additional evidence of the human ability for social chemosignaling: Producing and sending body odors that represent self-worth that is fundamental to social interaction.

It is important to note that individuals are able to hide their temporal or chronic emotional states. For instance, making a poker face or speaking in a steady voice to mask personal feelings of insecurity or to increase perceived dominance [29,30]. More relevant for the present purpose, people might manipulate the odors they emit by applying perfumes, i.e., explicitly manipulating the *cue*. The practice of using perfume ("self-anointment") to mask body odor is

ancient and widespread [31], and is encountered even in non-human primates and other animals [32]. One of the drivers of self-anointment could be to make a better impression on others [33]. People tend to choose perfumes that match well with their natural body odor, and perceivers of these smells also experience this self-chosen combination of fragrance and body odor as more pleasant than a random combination of fragrance and body odor [34]. In similar vein, Dalton, Mauté, Jean and Wilson [35] demonstrated that when participants were exposed to sweat from female participants who applied antiperspirant–blocking sweat excretion from the underarm–they rated the female participants as more confident, trustworthy, and competent in a stressful situation, compared to when they were exposed to their 'natural' sweat odor. This suggests that body odors with and without fragrance convey different social signals, and that it is possible to actively change the signal a sender emits [36,37]. Interestingly, fragrances people wear can also have an effect on their *own* behavior: when male participants wore perfume during recording of videos compared to a control condition, female observers rated these male contestants on videos as more dominant, confident, and attractive, traits closely related to self-esteem [38,39], even when female participants were unable to smell the males' body odors. Taken together, fragrances appear to change the impression we have on others in different ways.

In the present study, we examined for the first time whether self-esteem is a cue that can be inferred from body odor, and second, whether such signal changes social impressions. Furthermore, we explored how fragranced body spray applied by individuals affects these impressions. Our hypotheses were stated as follows (see https://osf.io/s9xpq/ for the pre-registration document):

1. If self-esteem is cued in body odor, exposure to sweat from male sender participants varying in self-esteem will modulate social judgments related to self-esteem, in female perceivers: exposure to sweat from low self-esteem male sender will cause lower perceived self-esteem compared to exposure to sweat from high self-esteem male senders.

2. If fragrance masks low self-esteem, sweat obtained from male senders with low self-esteem treated with fragrance will lead to higher perceived self-esteem in female perceivers.

These hypotheses were studied using a state-of-the-field sender-perceiver paradigm [e.g., 21,33,40,41], where first, sweat samples were collected from two groups of males, differing in self-esteem and either wearing fragrance or not, and second, where these samples were presented to a group of female perceivers.

## Methods

### Participants

**For the sender study**, a total of 35 healthy male participants were invited to participate in the study. These were selected from a large sample (*n* = 216) of males in the appropriate age range (i.e., ranging 18–35), based on their self-reported self-esteem (as measured using the Dutch version of the Rosenberg Self-Esteem Scale, RSES [42]), so that half of the participants scored in the lower quartile (i.e., had low self-esteem), and the other half scored in the upper quartile of the population (i.e., had high self-esteem). The RSES is a short questionnaire with 5 positive phrased and 5 negative phrased statements on self-esteem, answered using a 4-point agreement scale (ranging 'strongly agree'–'strongly disagree'). An example item is "On the whole, I am satisfied with myself". The Dutch version of the RSES has attested good validity and internal consistency [42]. Population quartile values of the RSES were determined by comparing larger scale population studies on self-esteem (See Supplementary Materials S2 in S1 File for

an extensive description of the demographics in this sample including population study comparison). The selection criteria were being male in the appropriate age range (18–35) with no concurrent psychological disorders.

Invited participants completed the RSES a second time to check consistency of the measurement. Of the 35 invited participants, two participants did not show up for the second sweat donation session, and a third participant was excluded because of very inconsistent values on the RSES (e.g., one participant scored 14 during initial screening vs. 28 during the information session). We initially aimed to obtain samples from 32 male participants, and invited more participants to account for potential dropouts. This sample size of 32 sender participants was based on what was necessary for the perceiver study: per fragrance condition, a sender participant provided 2 sweat pads, one for each arm, that were cut into 8 pieces. Stimuli were composed of 4 different pieces from 4 donor participants in the same condition and group. Thus, 16 sender participants in each group would provide enough sweat pads for ($\frac{2*8}{4} * 16 =$) 64 perceiver participants. See Fig 1 for a visualization of the procedure of the sender study.

Of these 32 participants, 16 participants ($M_{age}$ = 26; $SD_{age}$ = 5.4) scored low on self-esteem ($M_{RSES}$ = 16.3; $SD_{RSES}$ = 2.6) compared to the population average (i.e., for men, $M_{RSES}$ = 22.4; $SD_{RSES}$ = 6.1, $N$ = 242; [43]). The remaining 16 participants ($M_{age}$ = 27; $SD_{age}$ = 5.1) scored high on self-esteem ($M_{RSES}$ = 26.9; $SD_{RSES}$ = 1.0).

During the sender study, additional measures of state self-esteem were collected during both fragrance conditions: the State Self-Esteem Scale [44], a measure of signature size [45,46], and the Name Initial Preference Task [47–49]. We did not find an effect of fragranced body

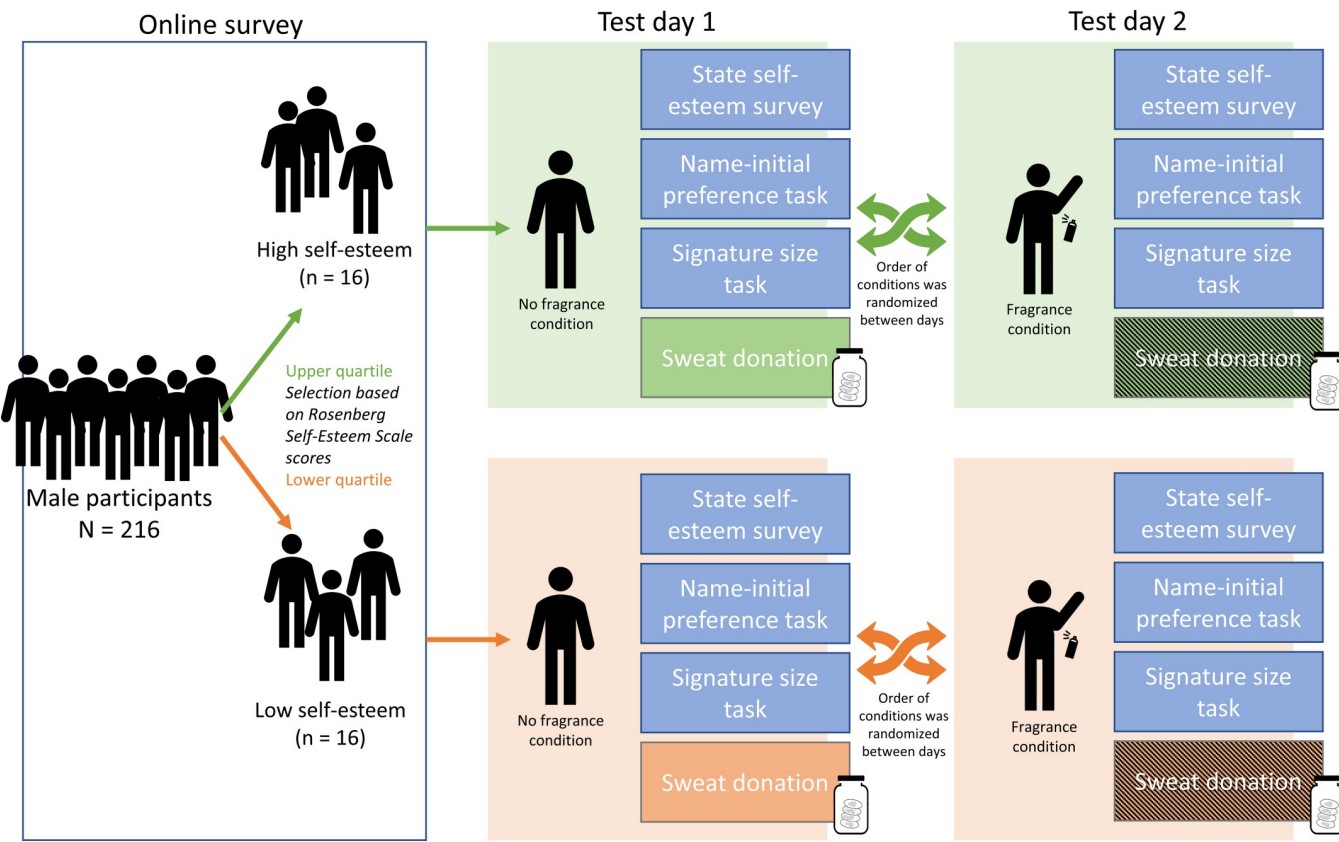

**Fig 1. Flow-chart of participant selection and procedure of the sender study.**

spray on self-esteem in these measures. These are reported in full in the Supplementary materials S2 in S1 File.

**In the perceiver study**, 62 heterosexual healthy female participants were recruited. Sample size was determined through a power analysis (using G*Power 3.1 [50]) for within subjects ANOVA (power = .8, α = .05). The effect size was based on similar research on the effect of sweat with or without body spray applied on the judgment of perceived stress and the closely to self-esteem related construct of confidence [35,51]. The effects in this study corresponded to $d = .41$ for the difference between untreated stress sweat and stress sweat treated with an antiperspirant on confidence ratings, and $d = .28$ for the difference in overall social judgements between the two sweat conditions. These effect sizes were pooled ($d = .35$) and converted to eta$^2$ ($\eta^2 = .0297$), yielding $n = 46$ as group size sufficiently powered to find a within-subjects effect. To cope with potential drop-outs, 66 female participants were aimed for, of which 62 were actually tested. Participants were screened, recruited and tested by an external agency. See Fig 2 for a visualization of the procedure for the perceiver study.

Perceiver participants were recruited with an age-range of 18–35, with a mean age of 24.6 years ($SD$ = 4.71). Perceivers were included based on their self-reported normal olfactory functioning, and completed a three Sniffin' stick normosmia test to confirm normal sense of smell [52]. Hummel, Pfetzing and Lötsch [52] report that when using 3 sniffin' sticks, a score of zero correctly identified smells sensitively indicates olfactory dysfunction. Five participants scored one out of three correctly identified smells, all other participants had either two (n = 16) or three (n = 41) correct. None of the participants was pregnant. Participants reported their use of hormonal contraceptives and were asked to estimate the number of days since most recent menstruation.

## Stimuli

**Body odors.** Body odors were collected following previously established sweat collection protocols [21,40]. See Supplementary Materials S2 in S1 File for a detailed description of the donation procedure. In brief, sender participants came to the lab twice, once to donate sweat without body spray, and once with body spray applied under the underarm. Participants followed a protocol starting 3 days before the donation day, involving shaving instructions, personal care product restrictions, food restrictions, and completion of a diet diary. Sweat (including fragrance in that respective condition) was collected on cotton pads, and immediately vacuum-sealed and frozen at -20C.

After the collection phase was completed, four different types of sweat stimuli were made by pooling sweat together from sender participants in the four different conditions: from the

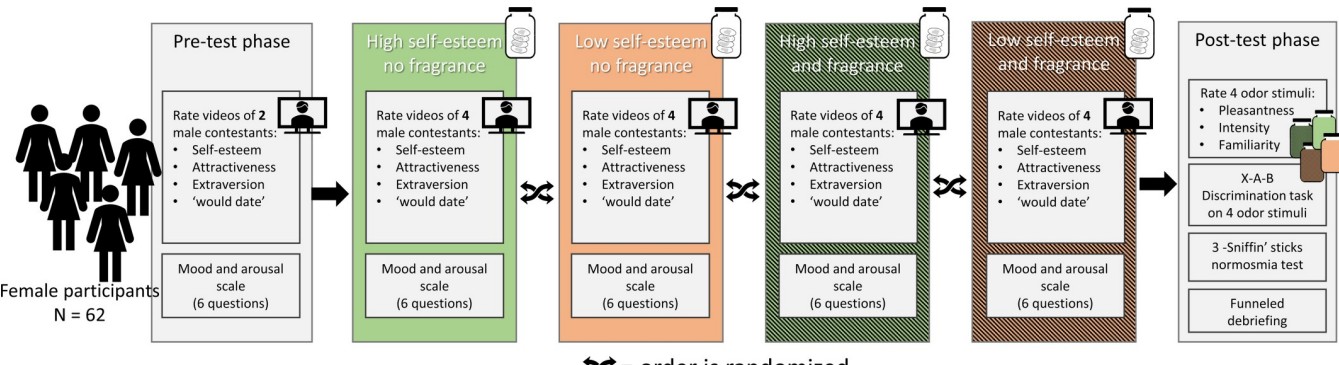

**Fig 2. Visualization of the procedure for the perceiver study.**

sender participants high in self-esteem and low in self-esteem, and from the same participants wearing fragrance or not wearing fragrance. Each sweat pad was cut in 8 pieces, and each stimulus composed of 4 pieces from different senders in the same self-esteem group and body spray condition. Stimuli were prepared in opaque 250ml food grade PE jars and stored frozen until test.

### Perceiver tasks

**Odor discrimination and qualitative judgment.** A two-alternative-forced-choice (2AFC) discrimination task was conducted to establish whether perceivers could discriminate between stimuli from senders high or low on self-esteem. Perceivers smelled one of the four different smell stimuli, and then smelled two smell stimuli, one of these being the same as the first. Participants had to indicate which of the two was the same as the first. Each smell was used as a target three times, with balanced alternative answer options, all presented in randomized order. Scores were added up for each smell (ranging 0 to 3, with 3 representing a perfect score). After the discrimination task participants were asked to rate each stimulus on pleasantness (a Likert scale ranging -4 'very unpleasant' to 4 'very pleasant'), intensity (ranging 0 'I don't smell anything' to 7 'strongest I ever smelled'), and familiarity (ranging 0 'not at all familiar' to 7 'very familiar').

**Psychological judgments.** A task was designed to elicit affective psychological judgements in a setting in which self-esteem and olfaction plays a role, i.e., interpersonal romantic interaction. Short videos showing interactions between a male contestant and a female contestant going on a first date were used as relevant test material, as self-esteem and body odor may modulate these interactions in an implicit manner. eighteen videos from a television dating show were selected, based on a pilot study (see the Supplementary table S3 in S1 File for a list of the videos used, and Supplementary materials S5 for a report on the pilot study in S1 File). Videos were selected portraying male contestants that were in the middle ranges of rated self-esteem, so that effect of sweat odor could or fragrance stimuli in either direction could be assessed. The videos were presented as embedded video with a width of 560 pixels and height of 315 pixels. All videos were played without sound (see Supplementary materials S5 in S1 File for a discussion on this), and lasted for approximately 30 seconds.

After watching each video, participants were asked to rate the male contestant on three characteristics ('given your impression of the man, how would you rate the man in the video on:', or *'Afgaande op jouw indruk van de man, in hoeverre vond je de man in het filmpje:'*, in Dutch), i.e., self-esteem *('zelfverzekerd'* in Dutch), attractiveness *('aantrekkelijk')*, and extraversion *('extravert' in Dutch)*, using a 0–100 visual analog scale ranging "not at all" to "very much".

As a brief training, before the odor conditions, participants watched two videos with male contestants that in the pilot-test were rated respectively higher and lower than the stimulus materials used during the conditions, to implicitly anchor participants in their choice and to make them familiar with the task. These videos are marked A and B in the Supplementary table S3 in S1 File. Then, the actual experiment started. Four trials with different scenes were used for each of the four smell conditions, making a total of 18 trials for the entire procedure (training and experiment–see S1 File). Scenes were presented in counter-balanced fashion using a Latin square randomized design, ensuring that every participant watched a different set of videos in each sweat type condition. The scores on each of the three questions were averaged over the four different videos in the same condition and taken as dependent measures.

**Mood and arousal.** Mood and arousal was measured using six semantic differentiation scale questions after the statement "I would like to know how you feel at this moment. Could

you indicate below whether how you feel matches with:” (“*graag zou ik willen weten hoe je je op dit moment voelt. Kun je hieronder aangeven in hoeverre je gevoel overeenkomt met:*”), with the pairs: good-bad (“*goed-slecht*”); sad-happy (“*bedroefd-vrolijk*”); pleased-displeased (“*tevreden-ontevreden*”); calm-excited (“*kalm-opgewonden*”); tired-energetic (“*vermoeid-energiek*”); sedate-aroused (“*rustig-opgewekt*”). People indicated their response by selecting their answer on a ten-point scale. The first three questions reflected mood, and the latter three questions measure arousal. These questions were taken from Aarts and Dijksterhuis [53], who based it on the Affect-Arousal Scale from Salovey and Birnbaum [54*]*. The two sets (reflecting mood and arousal respectively) of three questions were averaged per condition/baseline, scores for the four conditions were subtracted from a baseline measure taken before onset of the first odor trial to reflect change from baseline.

## Procedure

All methods were carried out in accordance with the declaration of Helsinki, and the Netherlands code of conduct for research integrity. The study was approved by the Ethics Committee of the Faculty of Social and Behavioral Sciences of Utrecht University, and filed under number FETC18-076. Written informed consent was obtained from all participants, who were all over 18 years old. All participants were instructed and tested in Dutch.

After signing the informed consent, perceiver participants were seated in a soundproof cubicle, with their head mounted in a fixed chinrest (see for detailed descriptions of the *sender* study procedure Supplemental materials S2 in S1 File, for a visual overview see Fig 1). They first did two practice trials of the video rating task with videos portraying a dating situation different from the videos used in the odor conditions, including the mood-arousal questions, to establish their mood-arousal baseline. They were then instructed to knock on the door to alert the experimenter, who would place the first odor stimulus. The four stimulus jars for each participant were thawed 30 minutes before the start of the experiment. Just before the onset of the condition at hand, the lid was removed, and the 250ml PE container for that condition was fixed in a clamp attached to the chin-rest at a fixed distance of 2 cm from their nose.

After every four trials of the rating task within one condition, a screenshot from one of the four previously presented videos was presented in counter-balanced fashion using a Latin square design. Participants were asked to answer the question how much they agreed with the statement “I would date this male” on a 7 point scale ranging “not at all” to “very much”. Data analysis from this question is presented in the Supplementary materials S4 in S1 File. After this, they rated their mood and arousal using the Mood-Arousal questionnaire. See Fig 2 for a visualization of the procedure of the perceiver study.

After going through all 4 odor conditions, they moved to a different location in the lab, and performed the discrimination task and gave qualitative judgments (i.e., pleasantness, intensity and familiarity) for each of the 4 stimuli in a randomized order. Finally, participants were tested for their normal olfactory functioning.

After their participation, participants were debriefed by means of a funneled interview. Most participants answered that they had an idea about the type of stimuli used. Several participants answered they thought the study was to test the effectiveness of a new type of deodorant, but none of the participants guessed the research question or hypotheses correctly.

## Design and statistical analyses

The perceiver study follows a within-subject design with the presentation order of the conditions counterbalanced (see Fig 2). Both experimenter and participant were blind to the odor condition. The dependent measures were analyzed by means of fully within participant 2*2

repeated measures ANOVAs, with Sender type (low vs. high self-esteem) and Fragrance (no vs. yes) as within participant factors. Since ANOVA is relatively robust against violations of normality, only in cases of severe violations appropriate transformations were applied [55]. Pairwise comparisons to follow up ANOVA tests were Bonferroni corrected.

Outliers were replaced following the median-absolute-deviation (MAD; [56]) outlier procedure, meaning that outliers, i.e., scores > 2.5 times the average median (i.e., the median absolute deviation) above or below the median score were replaced by scores one unit above or below the most extreme score still within 2.5 times the average median.

The main analyses were informally pre-registered by means of a time stamped, internally circulated PDF document (see Supplementary materials S1 in S1 File).

We also report Bayesian repeated measure analyses as an additional source of information about the evidence for the alternative hypothesis versus the null-hypothesis given the data. A uniform prior was used. Bayes factors for null versus the alternative ($BF_{10}$) or vice-versa ($BF_{01}$) are reported depending on whichever is > 1. Main analyses were performed in IBM SPSS Statistics, and Bayes factors were calculated using jamovi [57–59].

## Results

### Data statement

All data of this project, including syntax, can be found at https://osf.io/xwtnq.

### Discrimination task

First, we analyzed the data from the discrimination task, to test whether participants were able to discriminate between high self-esteem and low self-esteem sweat (see Table 1). The total numbers of correct answers for high self-esteem sweat versus low self-esteem sweat trials were set against the total amount of trials (i.e. the ratio of correct answers). Sensitivity ($d$'; [60]) was calculated and tested against chance level, i.e., (0), for the relevant trials.

Participants correctly discriminated low self-esteem sweat from high self-esteem sweat in 61.2% of the trials, which was significantly above chance ($p = .017$). For sweat masked with fragrance, participants were unable to discriminate between low and high self-esteem sweat, with performance around chance level (i.e., 52.8% correct, $p = .533$). To get a sense of the difference between pure sweat and sweat with fragrance, participants correctly discriminated between sweat from the same senders with and without fragrance in 94.8% of the cases ($p < .001$).

### Odor ratings

2-way ANOVAs with Sender type (low vs. high self-esteem) and Fragrance (no vs. yes) as within participant factors were performed on the ratings of pleasantness, intensity, and familiarity (see Table 2 for summary statistics). Normality tests indicated that the assumption of normality was violated for every condition (all $ps < .02$). However, no clear pattern of

**Table 1. Sensitivity values ($d$') and summary statistics for discrimination between sweat from low and high self-esteem sender participants.**

|  | % correct | Sensitivity ($d$') | SE | 95% Confidence interval for $d$' | $p$-value for $t$-test against chance |
|---|---|---|---|---|---|
| Low-No vs High-No | 61.2 | .40 | .12 | .16–.64 | .017* |
| Low-Yes vs High-Yes | 52.8 | .33 | .14 | .11–.54 | .533 |
| Low-No vs Low-Yes | 94.8 | 2.30 | .19 | 1.93–2.68 | < .001* |

**Note**: 'Low' denotes sweat from low self-esteem men, whereas 'High' denotes sweat from high self-esteem men. 'No' denotes that no fragrance was used, whereas 'Yes' denotes fragrance was applied. To illustrate, 'Low-Yes' denotes a stimulus originating from men with low self-esteem, who had applied fragrance.

**Table 2. Summary statistics for perceivers' (n = 62) body odor ratings of pleasantness, intensity, familiarity, with and without fragrance.**

| | Low self-esteem senders | | High self-esteem senders | | |
| --- | --- | --- | --- | --- | --- |
| | Without fragrance *M (SD)* | With fragrance *M (SD)* | Without fragrance *M (SD)* | With fragrance *M (SD)* | Significant effects *p* < .05 |
| Pleasantness | -1.19 (1.54) | 1.62 (1.69) | -0.39 (1.48) | 2.00 (1.34) | Frag.; Send. |
| Intensity | 2.44 (1.67) | 4.11 (1.21) | 1.50 (1.21) | 4.32 (1.10) | Frag.; Send.; Frag*Send. |
| Familiarity | 2.63 (2.12) | 4.02 (1.75) | 2.15 (2.09) | 4.09 (1.75) | Frag. |

Note: Pleasantness was rated on a Likert scale ranging -4 to 4, and intensity and familiarity were rated on a 0 to 7 Likert scale. Abbreviations: *Frag.* Means a significant main effect of fragrance use. *Send.* Means a significant main effect of Sender type. *Int.* means a significant interaction between Fragrance use and Sender type.

abnormality emerged for the four conditions, making a single transformation that corrected the different shifts from normality difficult, and distributions looked *close* to normal. Relative normality was further confirmed by skewness values, that were between [-.721; 1.064] for all ratings, and by kurtosis ratings, that were between [-.1.209; .265], and were thus all between the cutoff values for normality of -1.96 and +1.96 [58]. It was decided to proceed with the ANOVAs as planned.

For pleasantness, there was a main effect for Sender type: $F(1, 61) = 14.46$, $p < .001$, $\eta_p^2 = .192$, $BF_{10} = 2.0$, providing evidence that sweat from higher self-esteem men was rated as being more pleasant, independent of fragrance application. There was also a main effect for Fragrance: $F(1, 61) = 126.07$, $p < .001$, $\eta_p^2 = .674$, $BF_{10} > 100$, providing strong evidence that sweat samples from men who wore fragrance were rated as more pleasant, independent of Sender type. There was no evidence for a Sender type by Fragrance interaction, $F(1, 61) = 2.06$, $p = .157$, $\eta_p^2 = .033$, $BF_{01} = 2.7$.

For intensity, there was a significant main effect for Sender type: $F(1, 61) = 3.23$, $p = .015$, $\eta_p^2 = .093$, $BF_{01} = 1.9$, suggesting that sweat from higher self-esteem men was perceived as being somewhat more intense. There was decisive evidence that Fragrance made the stimuli perceived as more intense: $F(1, 61) = 176.33$, $p < .001$, $\eta_p^2 = .743$, $BF_{10} > 100$. However, these effects should be interpreted in light of a Sender type by Fragrance interaction, $F(1, 61)$ 12.68, $p = .001$, $\eta_p^2 = .172$, $BF_{10} = 73.4$. Without fragrance, sweat from high self-esteem men was rated as significantly less intense compared to sweat from low self-esteem men, $p = .001$, $d = .64$. Fragrance application removed this effect, making both sweat types comparably intense, $p = .211$, $d = .18$ (means and standard deviations are reported in Table 2).

Finally, for familiarity, there was no main effect for Sender type: $F(1, 61) = 1.62$, $p = .208$, $\eta_p^2 = .026$, $BF_{01} = 5.2$. However, there was a main effect for Fragrance: $F(1, 61) = 32.16$, $p < .001$, $\eta_p^2 = .345$, $BF_{10} > 100$, indicating that sweat samples containing fragrance were rated as more familiar compared to no-fragrance samples. There was no significant Sender type by Fragrance interaction, $F(1, 61)$ 2.53, $p = .117$, $\eta_p^2 = .040$, $BF_{01} = 2.6$.

In line with the results from the discrimination task, these results suggest sweat from men with high self-esteem smells different, i.e., less intense and more pleasant, compared to sweat from men low on self-esteem. However, fragrance application masked this effect. Next, we investigated whether different types of body odors also impacted on the affective judgments of the perceivers.

## Psychological judgments

To further investigate the effects of low and high self-esteem sweat on perceiver judgements of self-esteem, attractiveness, and extraversion, of videos showing male dating show contestants different to the sender participants, 2-way ANOVAs were done, again with Sender type and Fragrance as within participant factors (see Fig 3).

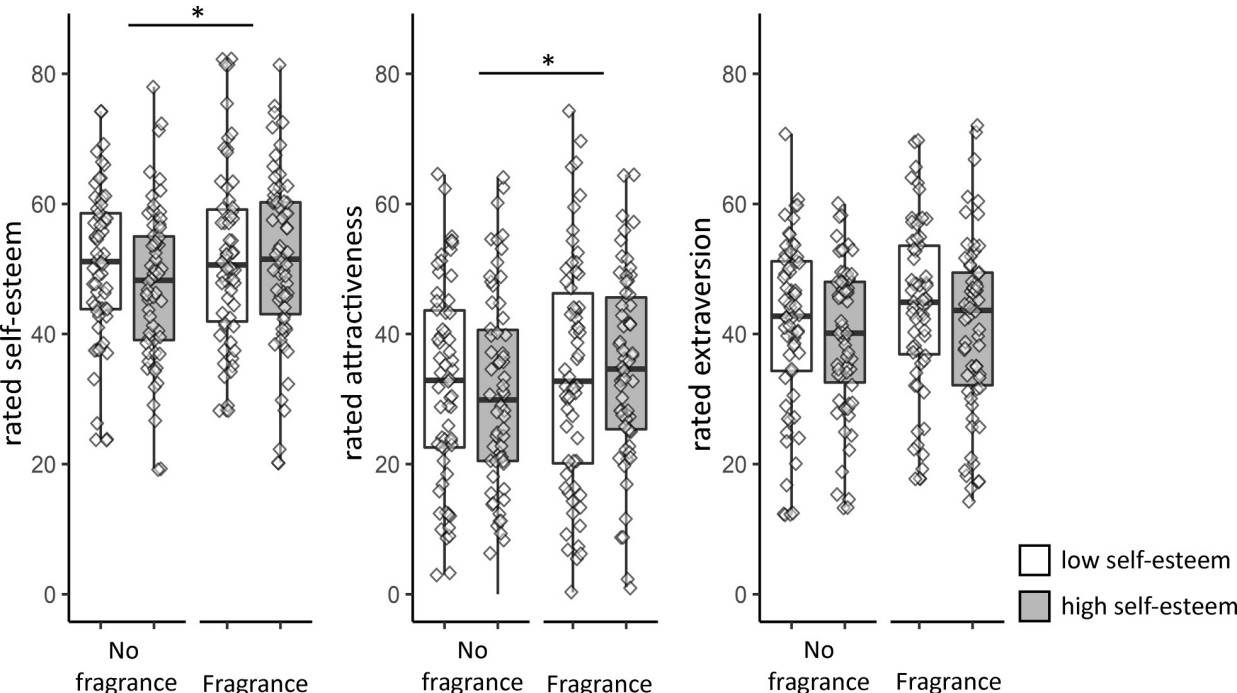

**Fig 3. Ratings for self-esteem, attractiveness and extraversion, under different stimulus conditions (sweat from low and high self-esteem donors, without and with fragrance).** There was a main effect of fragrance (denoted by an asterisks) for ratings of self-esteem and attractiveness, no other differences were significant. NB: Box-and-whisker plots display the median and 1st and 3rd quartiles ranges, and whiskers indicate range of the data. Diamonds denote individual data points. Perceiver participants rated videos that portrayed males (i.e., dating show contestants) who were not the sender participants (see the Psychological Judgments paragraph in the Methods section).

There was no main effect for Sender type on perceived ratings of self-esteem, $F(1,61) =$ 1.71, $p = .196$, $\eta_p^2 = .027$, with substantial evidence for the null-hypothesis of no effect, $BF_{01} =$ 3.4. However, there was a main effect for Fragrance, $F(1,61) = 4.04$, $p = .049$, $\eta_p^2 = .062$, $BF_{10} =$ 1.3, suggesting perceived self-esteem was higher when sweat of senders wearing fragrance was presented ($M = 51.65$, $SD = 11.45$) compared to sweat from the same senders without fragrance ($M = 49.03$, $SD = 8.85$). There was no significant Sender type by Fragrance interaction, $F(1,61)$ $= 0.89$, $p = .348$, $\eta_p^2 = .014$, with evidence for the null-hypothesis, $BF_{01} = 3.3$.

For perceived attractiveness, there was a similar pattern of results: the ANOVA showed no main effect for Sender type, $F(1,61) = .092$, $p = .763$, $\eta_p^2 = .001$, with substantial evidence that self-esteem levels of sweat senders did not impact attractiveness ratings during the perceiver phase, $BF_{01} = 7.1$. There was, however, evidence for a main effect for Fragrance, $F(1,61) =$ 5.53, $p = .022$, $\eta_p^2 = .083$, $BF_{10} = 3.0$, indicating that sweat collected while wearing fragrance ($M = 34.45$, $SD = 14.97$) resulted in higher attractiveness ratings during the perceiver phase than sweat without fragrance ($M = 31.58$, $SD = 14.28$). There was no significant Sender type by Fragrance interaction, $F(1,61) = 1.17$, $p = .284$, $\eta_p^2 = .019$, with evidence for the null-hypothesis of no interaction effect, $BF_{01} = 3.5$.

Turning to ratings of extraversion, there was no significant main effect of Sender type on rated extraversion, $F(1,61) = 3.56$, $p = .064$, $\eta_p^2 = .055$, $BF_{01} = 1.3$. There was also no main effect for Fragrance: $F(1,61) = 3.87$, $p = .054$, $\eta_p^2 = .060$, $BF_{01} = 1.6$, and substantial evidence for the null-hypothesis of no interaction between Sender type and fragrance on rated extraversion, $F(1,61) = 0.08$, $p = .784$, $\eta_p^2 = .001$, $BF_{01} = 5.4$.

The results of these analyses show perceivers rated men differently on their self-esteem and attractiveness, given their smell. However, the perceivers were not so much affected by the

type of sender who had donated the sweat, but were by whether or not the sender wore fragrance. We next investigated whether perceivers' self-reported mood was affected by the body odors they were exposed to.

## Mood and arousal

Change-from-baseline scores were calculated for the mood and arousal questionnaire scores. For mood, the analysis showed no effect of Sender type, $F$ (1,61) = .023, $p$ = .880, $\eta_p^2$ = .000, $BF_{01}$ = 7.3, no effect of Fragrance, $F$ (1,61) = .126, $p$ = .724, $\eta_p^2$ = .002, $BF_{01}$ = 6.8, and no significant interaction between Sender type and Fragrance, $F$ (1,61) = .611, $p$ = .437, $\eta_p^2$ = .010, $BF_{01}$ = 4.3.

The same analysis was repeated for arousal scores. There was no effect of Sender type, $F$ (1,61) = 1.41, $p$ = .239, $\eta_p^2$ = .023, $BF_{01}$ = 3.4, no effect of Fragrance, $F$ (1,61) = .037, $p$ = .847, $\eta_p^2$ = .001, $BF_{01}$ = 6.6, and no significant interaction, $F$ (1,61) = 2.29, $p$ = .135, $\eta_p^2$ = .036, $BF_{01}$ = 2.3.

These results suggest that although perceivers judge senders differently based on whether they wear fragrance or not, and can discriminate between the sweat of men with low or high self-esteem, these odors do not affect their mood or arousal state.

## Discussion

The present study investigated the social chemosignaling of self-esteem, according to which people with a high vs. low self-esteem produce perceivably different body odors that impact on social impressions related to self-esteem. Employing a sender-receiver paradigm, the data showed that female perceivers rated the body odors of male senders with high self-esteem as more pleasant and intense than body odors of male senders with low self-esteem, confirming the distinctiveness of body odor related to levels of self-esteem. However, these effects of chemosignaling of self-esteem did not alter ratings of male contestants who appeared in dating scenarios evaluations. Furthermore, applying fragrance to the body revealed that smelling sweat masked with scent led to higher ratings of potential dating partners.

Although low self-esteem body odor was perceived as less pleasant and more intense, this did not translate into negative social impressions. This suggests that feeling negative or positive about oneself, as indicated by self-reported self-esteem, does not necessarily translate into a social cue that affects others. One underlying mechanism for chemical signaling of personality traits is that they may be traced to socio-biological characteristics that in turn influence the composition of sweat [34]. For example, trait dominance has been suggested to be related to individual differences in testosterone levels [e.g., 61], and in turn, testosterone levels drive the neurobiochemical composition of sweat [cf. 62]. Consequently, dominant men might be more attractive to women in specific contexts [63]. However, self-esteem is often seen as a mediator of situational biochemical responses rather than as being under direct neurobiochemical control. For instance, self-esteem is related to cortisol response when people are socially challenged (i.e., social rejection; [64]). Thus, self-esteem might serve a social chemosignaling function when body odor is secreted under specific conditions. A possible way to follow this up could be to ask participants to donate sweat in an ostracism paradigm [65], to see whether social exclusion is signaled in sweat (e.g., in the form of stress) and perceptions of attractiveness, moderated by level of self-esteem.

The absence of a relationship between how people rate their own self-esteem and how their body odor influences ratings of self-esteem or attractiveness is nevertheless in line with previous findings on self-esteem and perceived attractiveness [66,67]. These studies suggest that while self-esteem is linked to how people judge their own attractiveness, this is of little

influence on how attractive others rate their appearance from videos. In addition, only small correlations between ratings of own self-esteem and peer-rated self-esteem have been demonstrated [68], and a weak cue of self-esteem in the sweat may have been obscured by other cues or the provided dating context. Nevertheless, we do find that body odors of men with high self-esteem are rated as more pleasant than those of men with low self-esteem. These results are somewhat contradictory. One possible explanation is that the receiver participants were implicitly distracted by a mismatch between the pooled sweat stimuli obtained from the male senders, and the contestants in the videos they were rating. In the current experiment, the videos portrayed contestants different from the body odor senders. The results may have been different if the videos of men portrayed the participants who donated the body odors, since that would align appearance with odor. Whether particular features of someone's appearance, e.g., one's eye or hair color or skin pigmentation, can be congruent or incongruent with someone's body odor is unknown, although this may be expected based on findings that very specific features such as relational status, age and gender are detectable in body odors [18,69,70]. Following the learning hypothesis of social chemosignaling [71], if particular appearance features are systematically paired with a particular chemical sweat profile, these features may be linked to that profile.

Another possible explanation for the absence of self-esteem body odor effect on social perception and behavior concerns the nature of our test setting. Previous research on social chemosignaling [21,23] mostly established effects of body odors on implicit measures of social evaluation or communication (such as facial electromyography or speeded classification of emotional facial expressions). In this case female participants provided explicit ratings of attractiveness and self-confidence. It is possible that social chemosignalling manifests at an implicit level of information processing as this rarely reaches consciousness [72]. Accordingly, although differences between high and low self-esteem were perceivable on explicit evaluative responses to the body odor itself (i.e. intensity, pleasantness and ability to discriminate), these difference did not affect the participants' social judgments in a social setting.

Our findings reveal that fragranced body odor can affect judgments of self-esteem and attractiveness. Dating show contestants were rated as more confident and more attractive when body odor with a fragrance was presented during the task compared to natural body odor. Fragrance, in this case, may be seen as communicative cue, since fragrance generally is applied intentionally, with a purpose. After all, fragrances have been designed carefully to contain molecules such as aldehydes, found in nature as part of the aromas flowers emit to attract pollinators like bees [73]. As such, it could be maintained that fragrances have been created precisely to mimic natural attractiveness cues. People tailor their impressions for particular audiences to show off their best self [29,34]. The present results show that this intentional signal, i.e., a fragrance applied on the body, is also picked up as such by perceivers. Previous studies indicate that using fragrance alters how people behave, affecting how others perceive their level of self-esteem and attractiveness in a test setting that did not allow for smelling the fragrance men had applied [74]. Similarly, previous work has also shown that using fragranced cosmetics can improve judgments of attractiveness and pleasantness based on body odors, and at the same time can partly obscure the perception of certain personality traits [33]. In line with these and other studies [37,74,75], the results suggest that just as people alter their voice, facial expression and face appearance to convey a particular impression to someone else [3,29], fragrances are effectively employed to alter one's chemical impression.

Whereas previous studies on chemosignaling (e.g., [76,77]) find mostly implicit effects of body odors, and no explicit perception effects, we found no implicit effects of odors, but only explicit perception effects of body odors. These perceivable differences in natural body odors did not affect the participants. In psychology, implicit measures are generally preferred as

evidence for a specific effect, as they are believed to provide more meaningful information than explicit judgments (e.g., [78,79]). Indeed, smells do not always receive overt attention, with many people not noticing but the most extreme changes in their surrounding *smellscape* [79,80]. At the same time, some people are much more aware of the smells in their surroundings. There are cultural differences in (social) odor awareness, but within country factors also play a large role [81,82]. For example wine and coffee experts are more aware of their sense of smell [83], and perfumers, women, and hunter-gatherers (see [84] for a review). This raises the question of what the effects of awareness on chemosignaling are. If one person barely takes notice of the strongest smells, while the next person is able to volitionally sniff out the subtlest differences in body odors (e.g., the smell of Parkinson's disease [85]), that raises questions whether there are individual differences in the effects of chemosignaling. An interesting alley for future studies would be to investigate whether particular social chemosignaling effects are only present when perceived unattentively (i.e., implicitly), or whether they lose, or—equally likely—gain, power when their presence is attentively noticed, for example by a highly aware and trained perfumer.

The present study was limited by a number of issues. Men and women differ in their ability to smell. Here, we have tested only female perceivers, smelling sweat from male senders. To be able to generalize to larger populations and to understand the underlying mechanisms of social chemosignaling, future research needs to include experimental groups including male perceivers, and female senders. The results suggest that female perceivers are able to discriminate between sweat from participants with high versus low self-esteem, but that this effect disappeared when the body odor was masked with fragrance. But interpreting these results should be done carefully, since sweat was pooled, every 8 of 64 stimuli contained sweat from the same individual, to mitigate individual differences in the sweat. However, particularly strong smelling sweat from one or a few individuals in one condition may have given a distinctive cue to many perceivers when discriminating the sweat, despite the efforts and hygiene standards applied in the sender protocol. Future studies with larger samples, collected using more ecological methods, may be able to overcome this limitation (cf. [86]). Another limitation is the use of the quick smell test [52], instead of a full test battery for olfactory function, to assess whether our perceiver participants were not impaired in their sense of smell (i.e., hyposmic or anosmic). By not excluding participants that did not score three correct identifications, it is possible that some of the participants in our perceiver experiment were hyposmic or even anosmic, even though they did not report this themselves. One methodological explanation for the absence of an effect of self-esteem status of the senders on rated self-esteem could lie in the fact that perceiver participants were not sensitive enough to detect these subtle differences in sweat. To investigate this possibility, the main analyses were repeated, including only the participants that scored 3 out of 3 correct (n = 41; the syntax for this analysis is available on https://osf.io/xwtnq/). This analysis yielded the same results, suggesting that in our sample, hyposmia does not explain the absence of an effect of sender self-esteem status on perceiver behavior.

In conclusion, we show that fragrance and self-esteem can affect sweat odor profiles of senders, in such a way that perceivers can pick up on those differences: men with low self-esteem smell different than men with high self-esteem. In addition, we show fragrance can have a substantial impact on how chemical cues are perceived by others (cf. [34]). In light of the widespread de-odorization and re-odorization in society, this provides important steps in understanding what the effect of fragrance on the non-verbal communicative function of body odors in social situations are. The results suggest that applying a fragranced olfactory layer onto one's body odor can mask sweat smells and alters the social impressions we make on others. The use of fragrances is pervasive, profound and widespread, and this study demonstrates that this is not without reason.

## Supporting information

**S1 File.**
(PDF)

## Acknowledgments

This study was sponsored by Unilever R&D. We are grateful to Philippe Valcasara and to Jasper de Groot for their contributions to this study. Recruitment and data collection was performed with the help of two external agencies. Data collection, processing and analysis was done by Utrecht University.

## Author Contributions

**Conceptualization:** Ilja Croijmans, Daniel Beetsma, Henk Aarts, Ilse Gortemaker.

**Data curation:** Ilja Croijmans, Daniel Beetsma.

**Formal analysis:** Ilja Croijmans.

**Funding acquisition:** Ilse Gortemaker, Monique Smeets.

**Investigation:** Ilja Croijmans, Daniel Beetsma.

**Methodology:** Ilja Croijmans, Daniel Beetsma, Henk Aarts, Monique Smeets.

**Project administration:** Ilja Croijmans.

**Software:** Daniel Beetsma.

**Supervision:** Monique Smeets.

**Visualization:** Ilja Croijmans.

**Writing – original draft:** Ilja Croijmans, Henk Aarts, Monique Smeets.

**Writing – review & editing:** Ilja Croijmans, Henk Aarts, Monique Smeets.

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
