## [Decision Letter · Decision Letter 0]

22 Feb 2021

PONE-D-20-36466

The smelly self: The role of self-esteem and fragrance in body odor and social impressions

PLOS ONE

Dear Dr. Croijmans,

Thank you for submitting your manuscript to PLOS ONE. The manuscript was reviewed by two independent reviewers who both believed that it has merit and is methodologically sounded. However, they both highlighted some changes that need to be addressed before proceeding for publication. Therefore, we invite you to submit a revised version of the manuscript that addresses the points raised during the review process.

We look forward to receiving your revised manuscript.

Kind regards,

Maria Serena Panasiti

Academic Editor

PLOS ONE

Journal Requirements:

3.Thank you for providing the following Funding Statement: 

"The research has been funded by Unilever. Researchers from Unilever provided input to the design of this study. Of the 5 authors listed, author Gortemaker was employed by Unilever during her contribution to the manuscript, and first author Smeets has a dual affiliation, working at Utrecht University and Unilever. Other authors are all affiliated with Utrecht University. The data reported in this paper have not been used for product development, and no intellectual property has been filed. Unilever authors were not involved in relevant statistical analyses. The design of the study, study implementation, and statistical analyses was led by authors from Utrecht University. Unilever would like to see these results published to help accelerate scientific progress in the area of sweat odor, and the impact of reodorization and deodorization. The funders had no role in study design, data collection and analysis, decision to publish, or preparation of the manuscript."

We note that one or more of the authors is affiliated with the funding organization, indicating the funder may have had some role in the design, data collection, analysis or preparation of your manuscript for publication; in other words, the funder played an indirect role through the participation of the co-authors.

If the funding organization did not play a role in the study design, data collection and analysis, decision to publish, or preparation of the manuscript and only provided financial support in the form of authors' salaries and/or research materials, please review your statements relating to the author contributions, and ensure you have specifically and accurately indicated the role(s) that these authors had in your study in the Author Contributions section of the online submission form. Please make any necessary amendments directly within this section of the online submission form.  Please also update your Funding Statement to include the following statement: “The funder provided support in the form of salaries for authors [insert relevant initials], but did not have any additional role in the study design, data collection and analysis, decision to publish, or preparation of the manuscript. The specific roles of these authors are articulated in the ‘author contributions’ section.”

If the funding organization did have an additional role, please state and explain that role within your Funding Statement.

Please also provide an updated Competing Interests Statement declaring this commercial affiliation along with any other relevant declarations relating to employment, consultancy, patents, products in development, or marketed products, etc.  

Reviewers' comments:

Reviewer's Responses to Questions

**Comments to the Author**

1. Is the manuscript technically sound, and do the data support the conclusions?

Reviewer #1: Yes

Reviewer #2: Yes

2. Has the statistical analysis been performed appropriately and rigorously? 

Reviewer #1: Yes

Reviewer #2: Yes

3. Have the authors made all data underlying the findings in their manuscript fully available?

Reviewer #1: Yes

Reviewer #2: Yes

4. Is the manuscript presented in an intelligible fashion and written in standard English?

Reviewer #1: Yes

Reviewer #2: No

5. Review Comments to the Author

Reviewer #1: In this study, the authors explored whether body odors can communicate information about one’s self-esteem, and the role of applied fragrance in this relationship. Even though high self-esteem sweat was rated more pleasant and less intense than low self-esteem sweat, there was no difference in perceived self-esteem and attractiveness of males in videos, hence explicit differences in body odor did not transfer to judgments of related person characteristics. When fragrance was added to the body odor, males were rated as having higher self-esteem and being more attractive.

The research question is well explained and related to the previous literature. Experiments have been conducted rigorously, with appropriate controls and sample sizes. I particularly appreciated the use of Bayesian analyses in support of frequentist analyses. The discussion is properly supported by the results. I have just few points that need to be addressed.

- I would suggest to change the title in order to give a sense of the findings. Right now, it sounds like body odors can communicate self-esteem

- At the end of the introduction, authors should present their hypotheses

- In the perceiver study, participants are women rating their impression of men. Did the authors control for the use of hormonal contraception and menstrual cycle? Previous literature has shown that these can affect the perception of body odors as well as social preferences so it might particularly relevant for the present study (Endevelt–Shapira et al., 2020; Derntl et al., 2013; Havlicek et al., 2005). If the authors collect this information it might be interested to see if the results are affected by these aspects.

- The procedure is a bit complicated, maybe a figure would help the reader to understand the different steps, odor stimuli and tasks.

Reviewer #2: Thank you for a possibility to review this submission. I believe that the topic is worth investigation and I found the study interesting and methodologically excellent. However, I also believe that the manuscript needs some work before it is published. My comments are mostly related to the writing style, missing details of the study and problems related to references (missing or erroneously applied). For example, the literature review in the introduction is good, but this section needs a bit of a reorganization and some shortening. One major change I would suggest is adding an analysis comparing baseline self-esteem assessments of the videos with the ratings performed during odor stimulation. Please find my detailed comments below.

Specific comments:

Abstract -

Please specify that the fragrance you mention in the abstract was a fragranced body odor.

„This suggests body odors associated with self-esteem have distinct perceivable features, but only fragrance affects the psychological impression someone makes.” - please rephrase, this sentence is not clear.

Introduction -

Please reorganize the introduction. The first paragraph (currently lines 29-38) should be placed after the paragraph on self-esteem (currently ending at line 63).

Paragraph on signal vs. cue (l. 64-76) is overly complicated and not entirely necessary. Please remove it (you can shorten the content and add it to the next paragraph).

The Sorokowska et al reference in line 77, Croy et al reference in line 86 and Roberts et al reference in line 123 are not well-matched to the content. Sorokowska et al study should be discussed in the paragraph on fragrances.

In the summary of the introduction „ In the present study, we examined for the first time whether self-esteem is a signal that can be inferred from body odor, and second, whether such signal changes social impressions.” - I believe you suggested that you referred to odors as cues?

Methods:

„This sample size of 32 sender participants was based on what was necessary for the perceiver study: 2 groups of 16 sender participants would provide enough sweat pads for 64 perceiver participants.” - please explain the rationale behind the selection of this sender sample size - how did you calculate that?

Please add a few words about the RSES in this section.

„During the sender study, additional measures of state self-esteem were collected.” - do you mean that the body odor donors were subject to one more measurement?

„Participants were screened” - screened for what?

„Five participants scored one out of three correctly identified smells” - were they eliminated from the study?

Please describe the videos in the „stimuli” section. Did the videos present the odor senders? You should also provide some references in the body odor section, showing that the procedure you used was valid and well-established.

Affective judgments are rather psychological judgments - please come up with a more exact name for this section.

“Affective judgments. A task was designed to elicit affective judgements in a setting in which self-esteem and olfaction plays a role, i.e., interpersonal romantic interaction.” - this is completely unclear.

Mood and arousal - can you provide more details on the original scale?

L. 240 - how were the odors presented?

Please provide a reference for the median-absolute-deviation (MAD) outlier procedure.

Results

“distributions looked close to normal” - this is oddly phrased. Maybe you can refer to skeweness and kurtosis here?

I understood from the methods that you have the baseline ratings of perceived self-esteem of the males in the videos? Please provide an additional analysis, comparing these ratings to the ratings performed in a presence of an odor.

Discussion

“ The observation that low self-esteem body odor was perceived as less pleasant and more intense, did not lead to negative social impressions, suggests that feeling negative about oneself, or positive, for that matter, does not necessarily produce a smell that serves a basic social chemosignaling function.” - please rephrase, this is very difficult to follow.

“One possible explanation is that the receiver participants were implicitly distracted by a mismatch between the pooled sweat stimuli, and the males in the videos they were rating.” - I do not understand this rationale.

“while explicit measures (such as attractiveness ratings) usually do not show differences.” - please provide some references here - I do not find this sentence convincing.

“Our findings do reveal fragranced body odor affects females’ judgments of self-esteem and attractiveness of males.” - please rephrase.

L. 459-476 Sorokowska et al 2016 also reported differences in conveyed impressions for natural vs. fragranced samples. Please discuss.

L. 476-490 - this paragraph is difficult to follow. Please rephrase or remove it.

6. PLOS authors have the option to publish the peer review history of their article (what does this mean?). If published, this will include your full peer review and any attached files.

Reviewer #1: No

Reviewer #2: No

---

## [Author Response · Author response to Decision Letter 0]

6 May 2021

Please see the 'response to reviewers' document attached for a point-by-point response to the reviewers' comments

---

## [Decision Letter · Decision Letter 1]

28 Jul 2021

PONE-D-20-36466R1

The role of fragrance and self-esteem in perception of body odors and impressions of others

PLOS ONE

Dear Dr.Croijmans,

Thank you for submitting your manuscript to PLOS ONE, as you will see from the comments below, both reviewers were happy with your revisions. One of the two reviewers still have few remaining comments, once you will carefully adress them I will be happy to accept the paper for pubblicaiton.

We look forward to receiving your revised manuscript.

Kind regards,

Maria Serena Panasiti

Academic Editor

PLOS ONE

Journal Requirements:

Reviewers' comments:

Reviewer's Responses to Questions

**Comments to the Author**

1. If the authors have adequately addressed your comments raised in a previous round of review and you feel that this manuscript is now acceptable for publication, you may indicate that here to bypass the “Comments to the Author” section, enter your conflict of interest statement in the “Confidential to Editor” section, and submit your "Accept" recommendation.

Reviewer #1: All comments have been addressed

Reviewer #2: (No Response)

2. Is the manuscript technically sound, and do the data support the conclusions?

Reviewer #1: Yes

Reviewer #2: Yes

3. Has the statistical analysis been performed appropriately and rigorously? 

Reviewer #1: Yes

Reviewer #2: Yes

4. Have the authors made all data underlying the findings in their manuscript fully available?

Reviewer #1: Yes

Reviewer #2: Yes

5. Is the manuscript presented in an intelligible fashion and written in standard English?

Reviewer #1: Yes

Reviewer #2: Yes

6. Review Comments to the Author

Reviewer #1: I thank the authors for taking the time to revise the manuscript according to previous comments. My concerns have been satisfied. I believe that the MS is now much stronger and worthy of publication in PlosOne.

Reviewer #2: This is my second review of the paper entitled “The role of fragrance and self-esteem in perception of body odors and impressions of others". I find the revised version of this manuscript improved and clearer and there are only a few issues I would suggest addressing.

Introduction:

p. 6 l. 111: “– blocking sweat excretion from the underarm, and thus any –“ – there is a word missing here.

p. 6 l. 116-120: the reference “(Craig Roberts et al., 2009)” is not relevant and incorrectly formatted.

p. 6 l. 122-123: “the psychological reasons underlying fragrance use, and the effects of this, has not been extensively investigated.” – This statement is not true, as there is a large body of literature showing multiple connections between psychological factors and fragrance use. Please reword or remove this sentence altogether.

Methods:

p. 8 l. 161: please clarify that you aimed to obtain 32 samples and invited more male senders to account for potential dropouts. Thank you for presenting the process in Figure 1, it is very useful.

p. 9 l. 193: “yielding n = 46 as group sizes” – group size per condition or a total sample size? Please verify and correct. The study is significantly underpowered if this is the required sample size per condition (I hope this is not the case).

p. 9 l. 201-203: you wrote that “Perceivers were included based on their self-reported normal olfactory functioning, and completed a three Sniffin’ stick normosmia test to confirm normal sense of smell (Hummel, Pfetzing, & Lötsch, 2010).” To confirm normal sense of smell based on the q-stics you would need to obtain the score of 3 for each participant, which means that many of your perceivers could have been hyposmic or even anosmic (self-reports are not very reliable, as shown by several publications). Please discuss this problem as a limitation of your study.

Results

p.15-16 l. 354-359: please move this part to the Discussion.

Table 2: Please modify this table to present the rated characteristics in consecutive rows, and columns low self-esteem donors (no deo-deo) followed by high self-esteem donors (no deo-deo) in columns. Perhaps you could also add a column “significant effects” and just briefly state, e.g., “Frag”, “Send”, “Frag*Send” This will help the reader grasp all your findings (I must admit I needed to read the whole section a few times to get the overall picture).

l. 409-448: Please write clearer that the psychological judgments refer to ratings of videos of men other than your senders.

Discussion

Please present the limitations of your study in a separate paragraph.

7. PLOS authors have the option to publish the peer review history of their article (what does this mean?). If published, this will include your full peer review and any attached files.

Reviewer #1: No

Reviewer #2: No

---

## [Author Response · Author response to Decision Letter 1]

16 Aug 2021

Please find the comments to the reviewer's points in the attached document titled 'Letter to reviewers R2 20210816.docx'

---

## [Decision Letter · Decision Letter 2]

6 Oct 2021

The role of fragrance and self-esteem in perception of body odors and impressions of others

PONE-D-20-36466R2

Dear Dr. Croijmans,

I am pleased to inform you that your manuscript has been judged scientifically suitable for publication and will be formally accepted for publication once it meets all outstanding technical requirements.

Please take care of making the last correction indicated by reviewer 2 in Table 2, you can do that during the proof process.

Kind regards,

Maria Serena Panasiti

Academic Editor

PLOS ONE

Additional Editor Comments (optional):

Reviewers' comments:

Reviewer's Responses to Questions

**Comments to the Author**

1. If the authors have adequately addressed your comments raised in a previous round of review and you feel that this manuscript is now acceptable for publication, you may indicate that here to bypass the “Comments to the Author” section, enter your conflict of interest statement in the “Confidential to Editor” section, and submit your "Accept" recommendation.

Reviewer #1: All comments have been addressed

Reviewer #2: All comments have been addressed

2. Is the manuscript technically sound, and do the data support the conclusions?

Reviewer #1: Yes

Reviewer #2: Yes

3. Has the statistical analysis been performed appropriately and rigorously? 

Reviewer #1: Yes

Reviewer #2: Yes

4. Have the authors made all data underlying the findings in their manuscript fully available?

Reviewer #1: Yes

Reviewer #2: Yes

5. Is the manuscript presented in an intelligible fashion and written in standard English?

Reviewer #1: Yes

Reviewer #2: Yes

6. Review Comments to the Author

Reviewer #1: (No Response)

Reviewer #2: Thank you for adressing all my previous comments. I am satisfied with the responses you provided and I endorse this manuscript for publication. I would only suggest replacing “int.” in Table 2 with “frag*send”.

7. PLOS authors have the option to publish the peer review history of their article (what does this mean?). If published, this will include your full peer review and any attached files.

Reviewer #1: No

Reviewer #2: No

---

## [Editor Report · Acceptance letter]

3 Nov 2021

PONE-D-20-36466R2 

The role of fragrance and self-esteem in perception of body odors and impressions of others 

Dear Dr. Croijmans:

I'm pleased to inform you that your manuscript has been deemed suitable for publication in PLOS ONE. Congratulations! Your manuscript is now with our production department. 

Kind regards, 

on behalf of

Dr. Maria Serena Panasiti 

Academic Editor

PLOS ONE